# A European International Multicentre Survey on the Current Practice of Perioperative Antibiotic Prophylaxis for Paediatric Liver Transplantations

**DOI:** 10.3390/antibiotics12020292

**Published:** 2023-02-01

**Authors:** Juliane Hauschild, Nora Bruns, Elke Lainka, Christian Dohna-Schwake

**Affiliations:** 1Department of Paediatrics I, University Hospital Essen, University of Duisburg-Essen, 45147 Essen, Germany; 2Department of Paediatrics II, University Hospital Essen, University of Duisburg-Essen, 45147 Essen, Germany

**Keywords:** paediatric liver transplantation, perioperative prophylaxis, antibiotics, antibiotic exposure, antimicrobials, infectious disease specialist, infection surveillance, infection prevention

## Abstract

(1) Background: Postoperative infections are major contributors of morbidity and mortality after paediatric liver transplantation (pLTX). Evidence and recommendations regarding the most effective antimicrobial strategy are lacking. (2) Results: Of 39 pLTX centres, 20 responded. Aminopenicillins plus ß-lactamase inhibitors were used by six (30%) and third generation cephalosporins by three (15%), with the remaining centres reporting heterogenous regimens. Broad-spectrum regimens were the standard in 10 (50%) of centres and less frequent in the 16 (80%) centres with an infectious disease specialist. The duration ranged mainly between 24–48 h and 3–5 days in the absence and 3–5 days or 6–10 days in the presence of risk factors. Strategies regarding antifungal, antiviral, adjunctive antimicrobial, and surveillance strategies varied widely. (3) Methods: This international multicentre survey endorsed by the European Liver Transplant Registry queried all European pLTX centres from the registry on their current practice of perioperative antibiotic prophylaxis and antimicrobial strategies via an online questionnaire. (4) Conclusions: This survey found great heterogeneity regarding all aspects of postoperative antimicrobial treatment, surveillance, and prevention of infections in European pLTX centres. Evidence-based recommendations are urgently needed to optimise antimicrobial strategies and reduce the spectrum and duration of antimicrobial exposure.

## 1. Introduction

The first paediatric liver transplantations (pLTX) were performed in 1963, becoming the treatment of choice for acute and chronic liver diseases that cannot sufficiently be treated otherwise [1]. Today, the reported one-year survival exceeds 85% [2,3,4]. The remaining causes of morbidity and mortality in children mainly comprise early post-operative complications such as non- and poor-function of the liver, thrombosis of the portal vein or hepatic artery, haemorrhage and infections [5,6,7]. Forty-seven to 82% of these infections derive from bacterial origin [8,9,10,11,12,13] with long surgery times, transfusion of blood products, medical immunosuppression and disturbance of the mucosal gut barrier as the main risk factors. In a large registry study that included 2291 patients, 38% experienced a bacterial or fungal infection within 30 days after transplantation, and 5.5% died as a consequence of infection [14]. In a single centre study including 345 transplantations, 127 cases of sepsis, 22 cases of severe sepsis and 41 cases of septic shock were reported within the postoperative paediatric intensive care unit stay. Within this study population, septic shock was the leading cause of death [13,14]. 

Given the major influence of bacterial infections on the postoperative course after pLTX, the importance of effective preventive strategies seems under-represented in the literature and in guidelines. Perioperative antibiotic prophylaxis strategies as one example of prophylactic measure differ between centres [9,12,13,15,16,17,18]. It aims to prevent mainly surgical site infections and bacteraemia, but optimal choice of antimicrobial agent and length of therapy remains uncertain.

This study investigates the different anti-infective strategies and applications of antibiotics of European paediatric liver transplantation centres based on an online-survey, aiming to give an overview on anti-infective prevention measures.

## 2. Results

### 2.1. Demographics of Participating Centres

Out of 39 pLTX centres that were contacted, 23 (59%) questionnaires were answered, of which 20 (87%) met the eligibility criteria for analysis. The self-reported region of the participating centres was Western Europe (10/20; 50%), Central Europe (4/20; 20%), Northern Europe (3/20; 15%), Southern Europe (2/20; 10%), and Eastern Europe (1/20; 5%). Most participating centers performed more than 10 pLTXs including a few high urgency pLTXs and a varying number of living related pLTXs *(*Figure 1a–c). Immediate postoperative care was performed on paediatric and mixed intensive care units (16/20 (80%) and 4/20 (20%), respectively). 

### 2.2. Immunosuppression

Seventeen (85%) centres used basiliximab for induction of immunosuppression. All centres included a calcineurin inhibitor (95% tacrolimus, 5% cyclosporine), and 13 (65%) centres used steroids as baseline immunosuppression with varying combinations of additional immunosuppressive substances (Table 1).

### 2.3. Antimicrobial Strategies

Standard perioperative antibiotic prophylaxis varied greatly, using an aminopenicillin plus beta lactamase inhibitor as most common choice (6/20; 30%) (Table 2). Eight centres (40%) used narrow spectrum antibiotics only, whereas 10 (50%) centres applied broad spectrum regimens. One centre reported that prophylaxis was tailored individually according to perioperative findings and another centre applied prophylaxis only to carriers of multidrug resistant bacteria. Antibiotics used to escalate treatment were mainly carbapenems, vancomycin, and ureidopenicillin plus beta lactamase inhibitor (Table 2).

The standard duration of antibiotic treatment varied across centres and was further adapted according to individual patients’ risk factors (Figure 2). Fourteen centres (14/20; 70%) gave detailed information about the considered risk factors. These were MDR colonisation (9/14; 64%), presence of an abdominal patch (8/14; 57%), postoperative course of c-reactive protein levels (7/14; 50%), postoperative course of procalcitonin levels (6/14; 43%), antibiotic treatment prior to transplantation (6/14; 43%), pre-existing conditions (4/14; 29%), indwelling central lines (3/14; 21%), ascites after surgery (2/14; 14%), patient’s age (1/14; 7%), length of hospitalisation prior to transplantation (1/14; 7%), and previous surgical procedures (1/14; 7%). When stratified for the annual number of transplantations, centres with lower numbers (≤20) and centres with higher numbers of transplant patients (>20) showed similar duration of prophylaxis, but centres with lower numbers used a narrow spectrum antibiotic more often (Appendix A). 

Antifungal prophylaxis in the absence specific risk factors was performed by 12 (60%) centres and included fluconazole, liposomal amphotericin B, micafungin, and caspofungin with varying risk factors triggering antifungal prophylaxis in the remaining centres (Appendix A). Cytomegalovirus (CMV) prophylaxis including aciclovir, ganciclovir or intravenous immunoglobulins was routinely administered to all patients in 7 of 17 (41%) centres and depended on additional risk factors in the remaining centres (Appendix A).

Infection surveillance strategies and non-pharmacological anti-infective measures varied greatly across centres (Appendix A). The majority of centres did not isolate the patients during the immediate postoperative course. Postoperative infectious management was mainly driven by teams of specialists that included a paediatric gastroenterologist in 14 (70%) centres, a paediatric infectious disease specialist in 10 (50%) centres, an infectious disease specialist in 6 (30%) centres, a paediatric intensive care specialist in 8 (40%) centres, a paediatric surgeon in 5 (25%) centres, an intensive care specialist in 3 (15%) centres, an anaesthetist in 2 (10%) centres and a surgeon, gastroenterologist or specialist for rational antibiotic therapy in one (5%) centre each.

### 2.4. MDR Prevalence and Perioperative Prophylaxis

The prevalence of MDR bacteria was low in the majority of centres (Appendix A). Among MRSA low prevalence-centres, 7 (69%) applied broad-spectrum antibiotics as prophylaxis (Table 3). The duration of prophylaxis ranged between 3–5 days in four of these centres and between 24–48 h in three centres. All centres with high MRSA prevalence used narrow-spectrum antibiotic prophylaxis. 

In the majority of centres with low ESBL prevalence (<20%) a narrow-spectrum prophylaxis (8 centres, 80%) was prescribed. The duration of the prophylaxis was limited to 24–48 h in four of these centres. Five of six centres with high ESBL prevalence used broad-spectrum antibiotics instead. Three of these centres limited the duration of the prophylaxis to 3–5 days. 

### 2.5. Availability of a (Paediatric) Infectious Disease Specialist and Perioperative Prophylaxis

Nine (56%) centres with involvement of an infectious disease specialist used narrow-spectrum antibiotics as prophylaxis. The duration of prophylaxis was limited to either 24–48 h or 3–5 days in 31.3% of the cases, respectively. Of the four centres without infectious disease specialist, three used broad-spectrum prophylaxis. 

## 3. Discussion

Children in the early phase after liver transplantation are at an increased risk for an infection, but evidence on optimal prevention strategies and current practice is limited. The results of this survey present an overview of anti-infective prevention strategies with a focus on perioperative antibiotic prophylaxis used in 20 pLTX centres across Europe. We observed striking differences between the centres especially regarding the choice and duration of antibiotic application. For example, the duration of perioperative prophylaxis ranged between an intraoperative single shot and 6–10 days. Similar differences were reported for antifungal, antiviral, non-pharmacological anti-infective measures and surveillance strategies. 

The observed differences reflect the under-represented topic of infection control and especially perioperative antimicrobial prophylaxis in the current literature. Reviews and state-of-the-art articles on paediatric liver transplantation neither include abstracts about infection control measures nor give any recommendations [19,20]. In a book chapter on early post-transplant management duration of postoperative prophylaxis is given for 5 days, either cefuroxime or cefazoline [21].

In the absence of guidelines or recommendations, the observed differences might be explained in part by centre specifics like prevalence of MDR pathogens, availability of infectious disease specialists and the clinical experience of the staff taking care of children after transplantation. Nevertheless, most centres in our study used longer antibiotic prophylaxis than the recommended standard in most of other major surgeries. This corresponds to the results of previous point prevalence studies, which reported prolonged antibiotic courses in critically ill children [22] and prolonged postoperative prophylaxis rates after major surgeries of up to 87% [23]. In this context, it seems necessary to point out that the potential harm of antibiotic is vast and must indispensably be outweighed against the potential benefits [24]. 

Very likely, the potential of reducing exposure to antimicrobial substances is not fully exploited. Early infectious complications after pLTX occurred in about 50% of cases in patients with antibiotic prophylaxis duration <48 h [12,13] and ≥48 h [9,15,16]. A pre-post design study on the implementation of standardized postoperative antimicrobial prophylaxis after pLTX achieved a reduction of broad-spectrum antibiotics covering mainly gram-negative bacteria for more than 48 h post-op from 77% to 44% and vancomycin use from 50% to 7% without an increase in adverse events [18]. As surgical site infection is one of the major contributors to complications in the early phase after pLTX, high quality prospective studies are needed to collect further evidence on the optimal duration of perioperative prophylaxis in order to optimize treatment effects and reduce harm from inadequate antibiotic exposure. Possibly, non-pharmacological or intraoperative measures and the improvement of surgery results carry the potential to further reduce surgical site infections [25]. 

Another striking heterogeneity of our survey was the choice of antibiotic substances used for perioperative prophylaxis. In the literature, similar heterogeneity has been reported with regimens ranging from carboxypenicillin plus ß-lactamase inhibitor to aminopenicillins plus third generation cephalosporins with or without metronidazole to ureidopenicillin plus ß-lactamase inhibitor with or without aminoglycoside [9,12,13,15,16,17,18]. In all cited studies, the rate of postoperative infections was reported at around 50%. The results of these studies suggest that the impact of specific antibiotic regimens on the development of postoperative infections after pLTX is limited. This limited impact may partially explain the different strategies between centres and at the same time carries the potential to optimise perioperative prophylaxis by narrowing the spectrum and duration of administered antibiotics to an acceptable minimum. 

An important barrier to narrowing the spectrum of perioperative prophylaxis after pLTX are MDR bacteria, as these pathogens constitute a clinically relevant cause of postoperative infections, sepsis, and septic shock [3,8,12,13,16,18]. The participants of our survey stated that they adapted antibiotic strategies according to the presence of MDR bacteria—centres with high prevalence of ESBL pathogens reported broader-spectrum regimens than low prevalence-centres. Apparently, prophylactic regimens are adapted according to local epidemiological considerations, which should also be taken into account when developing guidelines or recommendations in the future. Possibly, individually tailored perioperative prophylaxis is needed in MDR carriers.

Another factor that potentially influences the incidence and course of postoperative and surgical site infections in pLTX patients is the integration of infectious disease specialists in postoperative antimicrobial management. In this survey, 80% of the centres reported that they had support by infectious disease specialists. These centres were less likely to use broad-spectrum perioperative prophylaxis. Alongside optimised prescription of antimicrobials, further non-pharmacological measures of surveillance and prevention may play a role, such as routine cultures and swabs, isolation, and local antiseptic measures. The answers from our survey yielded great heterogeneity regarding these measures. In future prospective studies in the field of infectious complications after pLTX, these factors should be harmonized in order to rule them out as confounding effects of these measures and, if possible, gather evidence on their effectiveness. 

The major limitation of our study is the small number of participants, limiting in depth analyses on associations between different hospital characteristics and parameters of interest. Further, the survey was conducted anonymously with the self-reported region as only indicator of the geographical distribution of participating centres. This limitation is especially important to highlight as the prevalence of MDR bacteria varies very much across the regions, for ESBL from almost zero in Scandinavia to much higher percentages in Southern European countries [26]. Due to the anonymity, we cannot rule out double participation of a centre, even though we did not find duplicate answer profiles and no double naming of a hospital in the voluntary question. 

Nonetheless, this survey is an important contribution to understanding the current practice of perioperative antibiotic prophylaxis in Europe. The diversity of antibiotic and antimicrobial strategies and duration of prophylaxis we found is likely caused by local epidemiologic situations regarding MDR prevalence and further promoted by the lack of evidence-based recommendations. The availability of infectious disease specialists seems to foster narrow-spectrum perioperative prophylaxis, whereas high prevalence of ESBL pathogens was associated with broad-spectrum prophylaxis. The need to reduce harm from unwarranted, too long or too broad antibiotic treatment for the individual pLTX patient, critically ill children, and for society in general is high. This implies that prospective randomized controlled trials on the minimally necessary duration of perioperative prophylaxis, adequate substances, and effective additional measures after pLTX are urgently needed in order to optimise postoperative infectious care of these vulnerable patients. 

## 4. Materials and Methods

### 4.1. Study Design

The official contact persons of paediatric liver transplantation centres participating in the European Liver Transplant Registry (ELTR) were queried via email to answer an online survey (SurveyMonkey^®^) in December 2020 and in July 2021. Additionally, some centres were contacted personally by using official contact addresses from the hospitals’ websites. All questionnaires were filled in anonymously by the contact person from each centre with voluntary naming of the respective hospital.

The questionnaire was based on personal experience and the study of Vandecasteele on antimicrobial prophylaxis in adult liver transplantation [27]. It was reviewed, endorsed, and officially promoted by the ELTR. The questionnaire consisted of 30 questions including (i) the standard duration and type of antibiotic prophylaxis, the duration in patients with high risk for infection, and antibiotic choices in case of escalation, (ii) the annual number of total paediatric liver transplantations, living donor and high-urgency liver transplantations, (iii) the prevalence of multi-drug resistant bacteria (MDR), availability of infectious disease specialists, (iv) the strategies of immunosuppression, perioperative care and antiseptic measures, and (v) prophylactic antifungal and antiviral therapies (Appendix A). 

### 4.2. Data Analysis

Only questionnaires with at least the first four questions answered were eligible for analysis. Data are summarized as counts and frequencies. Because not all questions were answered by all participants, the denominator may vary and for that reason is given for each individual item. Some answers were categorized, e.g., the number of transplantations per year. All statistical analyses were performed with IBM SPSS Statistic for Windows, version 27 (Armonk, NY, USA). Figures were produced using Microsoft Office Excel for Mac Version 16.65 (Microsoft Corporation, Redmont, WA, USA).

### 4.3. Ethics Approval and Support

The European Liver Transplant Registry approved and supported the survey (acceptance letter 6 October 2020). The ethic committee of the medical faculty of Duisburg-Essen waived the need for ethic approval because no patient data were involved.

## 5. Conclusions

To conclude, this survey is the first to give an overview on current perioperative antibiotic prophylaxis after paediatric liver transplantation in Europe. We report inter-centre heterogeneity regarding all aspects of postoperative antimicrobial treatment, surveillance, and prevention of infections. The involvement of infectious disease specialists in postoperative management of infections was widespread and was associated with a higher proportion of narrow-spectrum perioperative prophylaxis. The results from this study imply that evidence-based recommendations are urgently needed in order to optimise pharmacological and non-pharmacological antimicrobial strategies and reduce exposure antimicrobials to the necessary minimum in terms of duration and spectrum. 

## Figures and Tables

**Figure 1 antibiotics-12-00292-f001:**
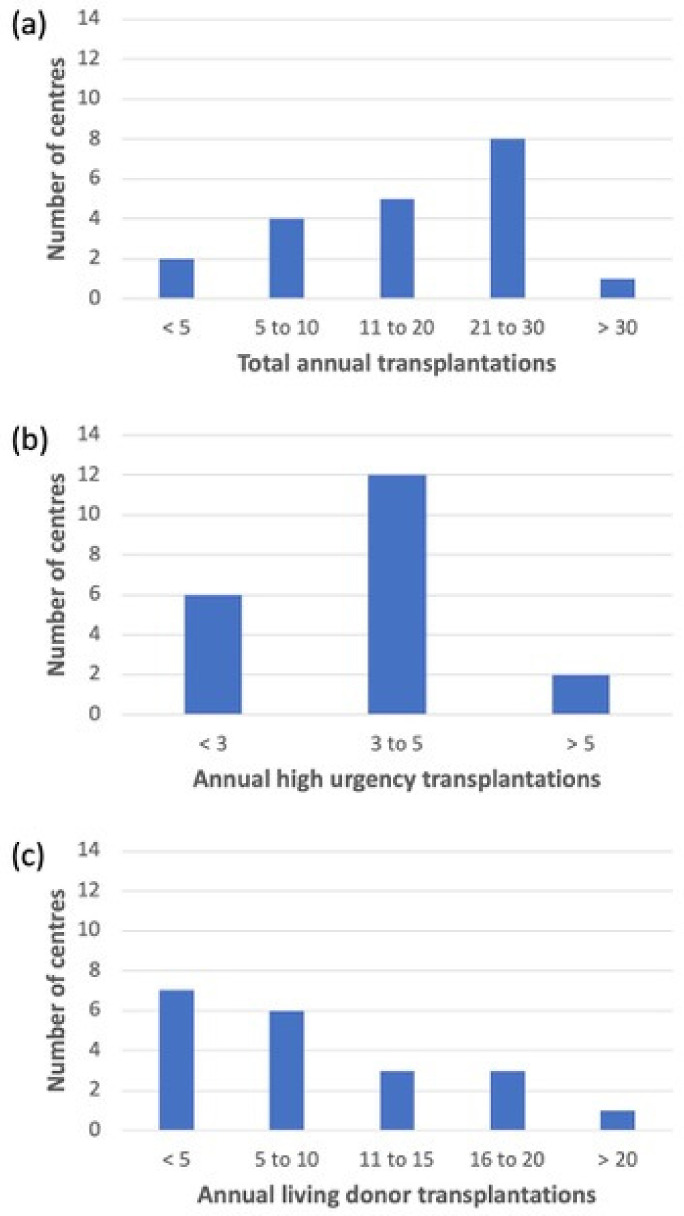
Annual frequency of paediatric liver transplantations in the participating centres. (**a**) Total number of annual transplantations. (**b**) Number of high urgency transplantations. (**c**) Living donor transplantations.

**Figure 2 antibiotics-12-00292-f002:**
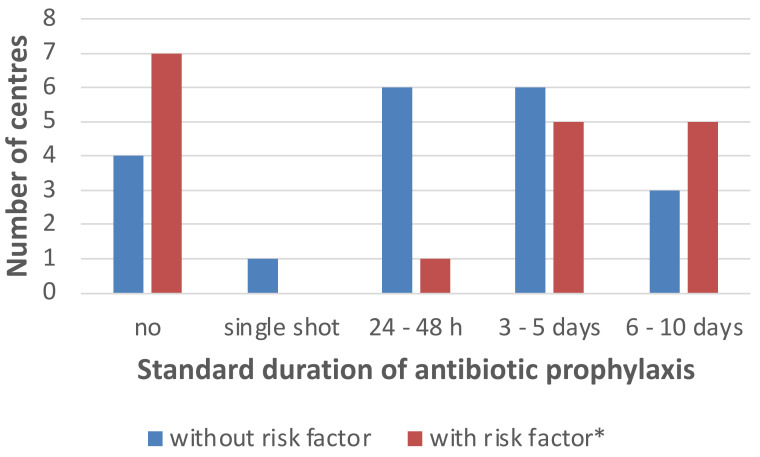
Duration of perioperative antibiotic prophylaxis, and according to additional risk factors in paediatric liver transplantation recipients. * Risk factors: defined by the treating physician/team, e.g., length of hospital stay, antibiotic treatment or intra-abdominal patch.

**Table 1 antibiotics-12-00292-t001:** Standard immunosuppression strategies in the first three weeks after paediatric liver transplantation in descending frequency.

n (%)	Steroid	Tacrolimus	CSA	MMF
10 (50%)				
6 (30%)				
2 (10%)				
1 (5%)				
1 (5%)				

CSA = Cyclosporine, MMF = mycophenolic mofetil. Grey boxes indicate the prescribed immunosuppression.

**Table 2 antibiotics-12-00292-t002:** Antibiotics for perioperative prophylaxis and for escalation therapy.

	Narrow Spectrum			Broad Spectrum								
n (%)	1st gen. Cephalosporin	Amino-Penicillin	Amino-Penicillin + BLI	3rd gen. Cephalosporin	4th gen. Cephalosporin	Ureidopenicillin + BLI	Carbapenem	Fluorquinolone	Glycopeptide	Amino-Glycoside	Colistin	Tigecycline
6 (30%)												
3 (15%)												
2 (10%)												
1 (5%)												
1 (5%)												
2 (10%)												
1 (5%)												
1 (5%)												
1 (5%)												
17 (85%)												
7 (35%)												
5 (25%)												
1 (5%)												
2 (10%)												
2 (10%)												
1 (5%)												
1 (5%)												
1 (5%)												

Green = use of narrow spectrum antibiotics only. Blue = Antibiotics used for escalation of treatment. Grey = use of either narrow and broad spectrum antibiotics or broad spectrum antibiotics only. BLI = Beta-lactamase inhibitor; gen. = generation.

**Table 3 antibiotics-12-00292-t003:** Association of prophylaxis spectrum with prevalence of multidrug resistance and availability of infectious disease specialists.

		n (%)	Narrow-Spectrum Prophylaxis n (%)	Broad-Spectrum Prophylaxis n (%)
(Paediatric) infectious disease specialist	Yes	16 (80%)	9 (56%)	7 (44%)
	No	4 (20%)	1 (25%)	3 (75%)
MRSA prevalence *	Low (<5%)	11 (69%)	4 (36%)	7 (64%)
	High (≥5%)	5 (31%)	5 (100%)	0 (0%)
ESBL prevalence *	Low (<20%)	10 (63%)	8 (80%)	2 (20%)
	High (≥20%)	6 (38%)	1 (17%)	5 (63%)

* Total number of answers n = 16.

## Data Availability

Data will be made available to any qualified researcher without undue reservation upon reasonable request.

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
