# Peer review of "A European International Multicentre Survey on the Current Practice of Perioperative Antibiotic Prophylaxis for Paediatric Liver Transplantations"

_antibiotics, 2023, doi:10.3390/antibiotics12020292_

Round 1
Reviewer 1 Report
My main concern relates to the presentation of the data. I strongly suggest to present data not only stratified according to number (%) of centers but also adding data on the number of patients per center.
Author Response
My main concern relates to the presentation of the data. I strongly suggest to present data not only stratified according to number (%) of centers but also adding data on the number of patients per center.
Reply: Dear reviewer, we appreciate your suggestion. We added a separate table as a supplement containing the information you requested, and added ana abstract in the manuscript.
Reviewer 2 Report
Dear Authors
I have reviewed your manuscript entitled "A European international multicentre survey on the current practice of perioperative antibiotic prophylaxis for pediatric liver transplantations." I find the manuscript clinically relevant to disseminate. However, the manuscript needs improvements before publication. Please see my comments below:
Introduction:
1. I am uncertain whether the information about complications and especially infections apply to adult patients undergoing liver transplantation or children. Please be more specific.
2. Are there any differences across Europe regarding what bacteria most commonly complicate the postoperative course of pediatric LTX? These may be important details before investigating the different anti-infective strategies and applications of antibiotics in European pediatric liver transplantation centers.
Results:
3. The first paragraph describing demographics should not duplicate information provided in Figure 1. I suggest changing it to "most participating centers performed more than 10 pLTXs including a few high urgency pLTXs...." or something similar and then refer to Figure 1.
4. My impression of the result section is that the reader receives too much unfiltered information from the questionnaire. I strongly recommend the authors shorten the number of figures and tables and produce a background table, preferably covering all the information needed.
5. As you have only a few responding centers, I do not find it suitable with Table 3, which reports relative risks with wide confidence intervals. This is because you need more events.
Discussion:
6. Overall, the discussion is well-written.
7. Could you provide information about perioperative antibiotic care in America and Asia?
Conclusion:
8. I find the wording of this sentence very negative: "We found great heterogeneity regarding all aspects of postoperative antimicrobial treatment...". It may be driven by "great heterogeneity." I would prefer "we report inter-center differences regarding postoperative antimicrobial...".
Supplementary:
For transparency, please include the questionnaire in the supplementary.
I wish the author the best of luck with this paper and future endeavors. I am happy to review a revision.
Author Response
Dear reviewer, thank you very much for your remarks which helped to improve the manuscript.
1. I am uncertain whether the information about complications and especially infections apply to adult patients undergoing liver transplantation or children. Please be more specific.
The information is related to children after transplantation. This is specified in the introduction.
2. Are there any differences across Europe regarding what bacteria most commonly complicate the postoperative course of pediatric LTX? These may be important details before investigating the different anti-infective strategies and applications of antibiotics in European pediatric liver transplantation centers.
As far as we know there are no specific investigations regarding the distribution of different bacteria complicating postoperative course. It is well known that the prevalence of MDR is much higher in Southern European countries. To account for these differences we asked for MRSA and ESBL prevalence, and related these data to different prophylaxis regimes.
3. The first paragraph describing demographics should not duplicate information provided in Figure 1. I suggest changing it to "most participating centers performed more than 10 pLTXs including a few high urgency pLTXs...." or something similar and then refer to Figure 1.
The first paragraph ws shortened according to your suggestion.
4. My impression of the result section is that the reader receives too much unfiltered information from the questionnaire. I strongly recommend the authors shorten the number of figures and tables and produce a background table, preferably covering all the information needed.
We are grateful for your comment and changed the results section. We have rearranged the results section, included more tables in the supplementary material and combined two tables into one in the manuscript.
5. As you have only a few responding centers, I do not find it suitable with Table 3, which reports relative risks with wide confidence intervals. This is because you need more events.
We have re-discussed this issue and we think that you are right. Relative risks are removed from the table and the manuscript.
6. Overall, the discussion is well-written.
Thank you!
7. Could you provide information about perioperative antibiotic care in America and Asia?
Thank you for the question, which is very hard to answer. We have again looked into the literature and medical books, but there is very limited data. We have added an abstract in the discussion on the fact that perioperative infection control measures are underrepresented in the current literature.
8. I find the wording of this sentence very negative: "We found great heterogeneity regarding all aspects of postoperative antimicrobial treatment...". It may be driven by "great heterogeneity." I would prefer "we report inter-center differences regarding postoperative antimicrobial...".
This was changed according your suggestion.
For transparency, please include the questionnaire in the supplementary.
Done.
Reviewer 3 Report
Thanks for this study that highlights the variations in clinical practice among livertx centers in Europe. Your work could be a good step to reach consensus among institutions providing livertx (a very specific treatment). Yet, there are several limitations to the study that limit the validity- with the anonymity of the respondents being the main one. You address this issue in the discussion, but some countries (or even some centers) could have been disproportionally included. I wonder if some centers use perioperative microbio samples to guide abx treatment- such as sometimes done in renal tx patients. Could the authors elaborate on that? Last, I would recommend a proof read by a native speaker to review for spelling issues.
Author Response
Dear reviewer, we appreciate your comments on our manuscript very much. Here is our response to your suggestions:
Yet, there are several limitations to the study that limit the validity- with the anonymity of the respondents being the main one. You address this issue in the discussion, but some countries (or even some centers) could have been disproportionally included. I wonder if some centers use perioperative microbio samples to guide abx treatment- such as sometimes done in renal tx patients. Could the authors elaborate on that? Last, I would recommend a proof read by a native speaker to review for spelling issues.
We have further adressed the issue of geographical distribution in the discussion.
In the supplementary section you can see that about 60% of the centres collected microbiological samples. Unfortunately, we did not ask if the results influenced the prophylaxis or escalation therapy.
We have made a native speaker review the manuscript.
Round 2
Reviewer 2 Report
Dear authors
You have made corrections to your manuscript in accordance with most of my comments. In my opinion, the manuscript has improved. However, the result section still includes very much information that would fit better in a table than the main text.
Author Response
Response to reviewer 2:
You have made corrections to your manuscript in accordance with most of my comments. In my opinion, the manuscript has improved. However, the result section still includes very much information that would fit better in a table than the main text.
- We agree that there is information that could be provided in a table. However, another reviewer criticised that there were already too many tables. For that reason, we shortened the text of results section by cutting out duplicate information that is also presented in tables, but did not create new ones. We hope that this compromise is feasible for you.
Reviewer 3 Report
Thanks for this revision.
I appreciate your efforts to improve the manuscript. Please provide the supplementary file for a proofread too; there are numerous spelling errors (e.g. leukocyten = leukocytes, nfectious = infectious, urin = urine, fluconazol = fluconazole, risc = risk)
Author Response
Response to reviewer 3:
I appreciate your efforts to improve the manuscript. Please provide the supplementary file for a proofread too; there are numerous spelling errors (e.g. leukocyten = leukocytes, nfectious = infectious, urin = urine, fluconazol = fluconazole, risc = risk)
- Thank you for the remarks. We corrected the spelling mistakes.